# Metabolic Inheritance and the Competition for Calories between Mother and Fetus

**DOI:** 10.3390/metabo13040545

**Published:** 2023-04-11

**Authors:** Edward Archer, Carl J. Lavie, Urska Dobersek, James O. Hill

**Affiliations:** 1EvolvingFX LLC, Fort Wayne, IN 46835, USA; 2Department of Cardiovascular Diseases, John Ochsner Heart & Vascular Institute Ochsner Clinical School—The University of Queensland School of Medicine, New Orleans, LA 70121, USA; 3Department of Psychology, University of Southern Indiana, Evansville, IN 47712, USA; 4Nutrition Obesity Research Center, University of Alabama at Birmingham, Birmingham, AL 35294, USA

**Keywords:** maternal, fetal, competition inheritance, metabolism, obesity, T2DM

## Abstract

During the prenatal period, maternal and fetal cells compete for calories and nutrients. To ensure the survival of the mother and development of the fetus, the prenatal hormonal milieu alters the competitive environment via metabolic perturbations (e.g., insulin resistance). These perturbations increase maternal caloric consumption and engender increments in both maternal fat mass and the number of calories captured by the fetus. However, a mother’s metabolic and behavioral phenotypes (e.g., physical activity levels) and her external environment (e.g., food availability) can asymmetrically impact the competitive milieu, leading to irreversible changes in pre- and post-natal development—as exhibited by stunting and obesity. Therefore, the interaction of maternal metabolism, behavior, and environment impact the competition for calories—which in turn creates a continuum of health trajectories in offspring. In sum, the inheritance of metabolic phenotypes offers a comprehensive and consilient explanation for much of the increase in obesity and T2DM over the past 50 years in human and non-human mammals.

## 1. Introduction

Competition is inherent in all biologic phenomena—from mating to metabolism [1,2,3,4,5]. Thus, it is the ‘struggle for existence’ that distinguishes mere physicochemical objects from organisms—molecules and mountains do not compete to survive and reproduce, but cells and people do. Therefore, *“nothing in biology makes sense except in light of competition”* [2].

This assertion, in concert with evolutionary theory, reveals that multi-celled organisms are the descendants of ancient single-cell creatures whose competitive success and subsequent reproductive fitness were improved by aggregation—the grouping of cells into coherent organisms [1]. Therefore, mammalian bodies can be viewed as ‘ecosystems’ comprising trillions of semi-autonomous cells competing for the nutrient-energy resources necessary for survival (‘calories’; glucose, fatty- and amino-acid molecules).

This conceptualization—that the body is a competitive ‘ecosystem’ of cells—offers a comprehensive framework for understanding the inheritance of metabolic phenotypes and subsequent development of obesity, type-2 diabetes mellitus (T2DM), and cardiovascular diseases (CVD) in human and nonhuman mammals. Yet, to grasp how the ‘*competition for calories*’ between mother and fetus impacts inheritance across species, it is necessary to understand the evolutionary foundation of the ‘*Competition Model of Metabolism*’ [1,2].

As such, this paper has three foci—presented in Section 2, Section 3 and Section 4. The first is a brief summary of the evolution of the ‘*competition for calories*’ in mammals [1,2]. The second is an explanation of how a mother’s environment, metabolism, and behavior impact metabolic inheritance and the subsequent development of metabolic diseases [6]. The third illustrates how ‘*maternal effects*’ (non-genetic mechanisms of inheritance [6,7,8,9] offer a comprehensive and consilient explanation for recent increments in the prevalence of inherited (pediatric/childhood) obesity and adolescent T2DM [1,6,10].

## 2. The Evolution of the Competition for Calories

The functional unit in biology is the cell [1,2,11]—and because all cells require nutrient-energy, the fundamental ‘problem’ in evolution was how a cell could acquire the ‘calories’ it needed for survival and reproduction. In the early stages of biologic evolution, this problem was ‘solved’ by chemical gradients and diffusion—if a single-celled creature was in the ‘right place at the right time’, nutrient-energy would simply diffuse across the cell membrane. Yet, as the population of primordial cells increased, the competition for resources intensified due to the zero-sum nature of nutrient-energy—‘calories’ are a finite resource because a molecule of glucose cannot be in two cells at the same time. As a result, over time, differences in cell phenotypes (physiology and behavior) led to differences in caloric acquisition with concomitant differential survival and reproduction.

Therefore, it can be argued that the competition for calories, in concert with the heritability of successful phenotypes, launched ‘*Evolution by Natural Selection*’ [5]. Nevertheless, the evolutionary potential of unicellular organisms was limited by a lack of specialization—single cells cannot multi-task efficiently (e.g., simultaneously move and capture calories). Thus, the ‘arrival’ of multi-celled organisms via the grouping of cells with different phenotypes into coherent, self-contained ‘ecosystems’ was inevitable. As such, without the ‘*competition for calories*’, Earth would be populated solely by single-celled creatures.

However, the evolution of multicellularity introduced two novel, existential ’problems’. First, by what mechanisms could the organism gauge and subsequently acquire the calories needed to support all the cells in its ecosystem? Second, given that specialized cells have unique phenotypes (physiology and behavior) and concomitant caloric requirements—by what mechanisms could each cell-type acquire the calories it needed while competing against every other cell?

Axiomatically, the problems of the acquisition and timely distribution of nutrient-energy were solved by all modern plants and animals—as evidenced by their continued existence. Yet, as detailed below, the evolution of mammals revealed one of Nature’s most elegant yet simple solutions [1,2].

### 2.1. The Mammalian ‘Ecosystem’

In mammals, the question of caloric acquisition was answered by the organismal (whole-body) behaviors of eating and drinking, whereas the question of the timely distribution of calories to each cell in the ecosystem was answered by the circulatory system in concert with cell-type-specific competitive behaviors (acquisition ‘strategies’).

### 2.2. Organismal Behavior: Consumption and ‘Effective Caloric Intake’

To meet the metabolic demands of all cells in their ecosystems, humans and other mammals developed sensorimotor-cells in the liver and brain that, when depleted of calories, initiate consumption (eating and drinking) [1,2]. These sensorimotor-cells are similar to other neuromuscular networks that coordinate sensation and bodily movement for survival (e.g., somatic reflexes and the ‘orienting response’ [12]). Because the sensorimotor-cells—like all cells in the body—compete for calories, their relative deprivation is indicative of the sufficiency of caloric intake for the entire organism. This simple but plastic mechanism ensures that when food is available, dietary consumption is sufficient for the maintenance of body mass and blood glucose levels.

To advance the understanding of the initiation of eating and drinking, we coined the term *“effective caloric intake”* to denote the number of calories required to inhibit consumption via the saturation of the sensorimotor-cells [1]. Specifically, when competition from other cells renders the ‘effective caloric intake’ insufficient, the sensorimotor-cells become depleted and consumption is initiated to reduce the deficit [1,2,13]. Conversely, when these cells are saturated, consumption is diminished.

There are several physiologic contexts in which the deprivation of the sensorimotor-cells occurs. These include negative energy balance (inadequate consumption relative to expenditure), *physiologic* and *pathologic* insulin resistance (IR), and excessive adipogenic partitioning (the capture, storage, and sequestration of calories in ‘fat-cells’, i.e., adipocytes). As detailed in later sections, *physiologic* IR occurs naturally as a result of increased growth-hormone (GH) secretion during puberty and pregnancy [6,14,15,16,17,18], whereas *pathologic* IR is a morbid condition induced via inadequate physical activity (PA) causing reductions in skeletal muscle-cell metabolism, insulin sensitivity, and glycogenesis—the storage of blood glucose as glycogen [1,2,10,13].

Importantly, all deficits of the sensorimotor-cells—independent of the cause—lead to increased consumption. However, because deficits caused by inadequate caloric consumption relative to expenditure (negative energy balance) threaten the survival of the organism, they generate levels of caloric intake sufficient to maintain body and fat mass. Conversely, deficits caused by IR or excessive adipogenic partitioning do not threaten the survival of the organism. As a result, these deficits lead to levels of consumption that engender positive energy balance (hyperphagia), with concomitant gains in body and fat mass.

As detailed in Section 3 and Section 4, the *physiologic* IR of pregnancy and concomitant overconsumption are essential for fetal growth and development, whereas *pathologic* IR perturbs the competition for calories between mother and fetus and irreversibly alters offspring metabolic phenotype and health trajectories.

### 2.3. Cell-Type-Specific Behavior

The survival of multi-celled organisms required that all cells in the ‘ecosystem’ capture the calories needed to meet the demands of their phenotype (physiology and behavior). This requirement led to the evolution of cell-type-specific competitive behaviors. Although all cell-types impact the competitive milieu by removing nutrient-energy molecules from the blood, skeletal muscle- and fat-cells are unique in their effects.

### 2.4. Skeletal Muscle-Cells

Humans and non-human mammals evolved in environments in which survival required voluminous amounts of bodily movement (physical activity, PA) despite uncertain food availability [19,20,21,22,23,24]. Given that the musculoskeletal system generates the PA necessary for survival and reproduction—acquiring food and water, avoidance of predators (fight/flight), and mating—it follows that skeletal muscle-cell metabolism is the most important and variable component of the competitive milieu. For example, skeletal muscle-cells are responsible for 44–55% of total daily energy expenditure whereas the brain is responsible for 20–25% [2,25,26,27,28,29,30,31].

To meet these demands, skeletal muscle-cells developed strategies that allow them to capture and exploit all major nutrient-energy molecules (e.g., glucose, fatty- and amino-acids). These strategies include insulin- and contraction-mediated glucose and fatty-acid metabolism. For example, during intense exercise and the post-prandial period (when fed), skeletal muscle-cells account for 70–85% of glucose uptake, whereas during low levels of PA and the post-absorptive period (when fasted), skeletal muscle-cells are a major site of fatty-acid oxidation [32,33].

As detailed elsewhere (see [1,2,6,13]), because physical movement directly impacts skeletal muscle-cell metabolism and the subsequent partitioning of calories to skeletal muscle-cells (e.g., glucose and fatty-acid storage), adequate levels of PA are required to maintain insulin sensitivity, and metabolic capacity and control. Consequently, insufficient PA engenders a loss of metabolic control (e.g., *pathologic* IR) that leads to hyperinsulinemia (high blood insulin levels), hyperphagia (overconsumption), and excessive adipogenic partitioning (the disproportionate capture, storage, and sequestration of calories in fat-cells).

As detailed in Section 3 and Section 4, given the role that skeletal muscle-cell metabolism plays in the maintenance of insulin sensitivity and metabolic control (e.g., glucose storage), it is not surprising that maternal PA directly alters fetal development and the inheritance of metabolic phenotypes.

### 2.5. Fat-Cells

From an evolutionary perspective, a fat-cell’s role in the mammalian ecosystem is to store calories during the post-prandial period (when fed) that can be used by other cell-types during the post-absorptive period (when fasted) [32]. As a result, fat-cells evolved unique insulin-dependent strategies that allow them to capture, store, and sequester (retain) calories well beyond organismal and cellular metabolic demands by increasing in size (hypertrophy) and number (hyperplasia) [2,13].

Specifically, as insulin levels rise during and immediately after meals, fat-cells remove glucose and fatty-acids (calories) from the blood. This causes them to increase in size (hypertrophy). Under conditions of neutral energy balance (calories ‘burned’ equal calories consumed), the stored calories are released to other cell-types during the post-absorptive period (e.g., over night) [32]. This reduces hypertrophy. Thus, fat-cells ‘grow’ when calories are consumed and ‘shrink’ when those calories are released and burned by other cell-types—primarily skeletal and cardiac muscle-cells [1,2,32,33,34,35,36,37].

However, under conditions of chronic positive energy balance (fewer calories ‘burned’ than consumed) and concomitant hyperinsulinemia, fat-cells increase in both size and number (hypertrophy and hyperplasia, respectively) [1,2,34,35,38,39,40]. The confluence of hyperinsulinemia and hyperplasia (increased number) allow fat-cells to capture, store, and retain an excessive number of calories after each meal. This chronic adipogenic partitioning decreases the ‘effective caloric intake’ of meals—which in turn increases consumption and engenders exponential increments in body and fat mass [1,2,10,13].

As detailed in Section 3 and Section 4, maternal insulin sensitivity, fat mass, and fetal fat-cell hyperplasia play decisive roles in the inheritance of metabolic phenotypes.

## 3. The Inheritance of Metabolic Phenotypes

Mammalian pregnancy—the development of a fetal ecosystem within the maternal ecosystem—necessitated the evolution of mechanisms to meet the caloric demands of both the mother and the developing fetus. However, because pregnancy evolved in prehistoric environments that necessitated voluminous amounts of PA despite uncertain food availability, the most critical determinant of appropriate fetal development is the balance between the number of calories consumed by the mother and those captured by maternal skeletal muscle-cells and the fetal ecosystem.

As detailed below, although the maternal–placental–fetal triad can withstand a wide range of perturbations, excessive or inadequate maternal PA can engender impoverished or over-nourished intrauterine environments that irreversibly alter pre- and post-natal development and the subsequent health trajectories of offspring.

### 3.1. ‘Normal’ Pregnancy and the ‘Competition for Calories’

During pregnancy, a mother’s cells compete for calories with the ever-growing population of fetal cells [1,6,13]. To ensure that the fetal ecosystem captures the calories needed for development, pregnancy alters the hormonal milieu (e.g., increased growth hormone (GH) secretion) [6,14,15]. Because GH reduces skeletal muscle-cell insulin sensitivity [18]—and the consequent storage of blood glucose—increased secretion causes *physiologic* IR [1,2,13,41,42].

This naturally occurring IR has several effects. First, higher levels of blood sugar stimulate maternal pancreatic beta-cells to increase insulin production. [Note: females who lack pancreatic beta-cell plasticity and cannot increase insulin production to meet the demands of pregnancy develop gestational diabetes mellitus (GDM)]. Second, the resulting hyperinsulinemia increases the competitive advantages of maternal fat-cells, which in turn, leads to excessive adipogenic partitioning. Third, the IR-induced partitioning reduces the flow of calories through the sensorimotor-cells, thereby diminishing the ‘effective caloric intake’ of meals. This stimulates maternal overconsumption (hyperphagia).

Together, as depicted in Figure 1, these metabolic perturbations lead to the gains in maternal body and fat mass, and the positive energy balance necessary for appropriate fetal growth and development [1,2,6,13].

### 3.2. Pathologic Pregnancies

#### 3.2.1. Intrauterine Growth Restriction

Intrauterine growth restriction (IUGR) occurs when a mother’s caloric consumption and nutrient-energy stores (e.g., body fat) cannot meet the demands imposed by both her cells and fetal development. In other words, the failure to create a consistent positive energy balance in the fetus inhibits normal development and leads to ‘small for gestational age’ (SGA) neonates.

For example, IUGR and SGA neonates are prevalent in subsistence farming communities because of high levels of maternal PA in concert with low food availability. To be precise, high levels of PA engender competitive advantages in maternal skeletal muscle-cells (e.g., elevated insulin sensitivity) that allow them to outcompete all other cells. Therefore, if a mother fails to consume calories sufficient to meet the demands of her cells and those of fetal development, the ensuing negative energy balance causes extreme fatigue in the mother, inadequate gestational weight gain (GWG), and an impoverished fetal ‘ecosystem’ with fewer muscle-, bone-, and organ-cells.

The reduction in bone-cells leads to reduced child and adult height (stunting), whereas reductions in skeletal muscle-cells causes lifelong decrements in muscle strength, PA, and metabolic capacity and control. Together, as depicted in Figure 2, these metabolic sequelae increase the risk of obesity, T2DM, and CVD as advanced by the ‘*Barker Hypothesis*’ [43,44,45,46] and the more comprehensive ‘*Maternal Resources Hypothesis*’ [6,10,47,48,49,50].

Importantly, IUGR occurs along a continuum of energy balance between mother and fetus. As such, pregnant females with low fat mass who consume too few calories relative to their PA levels will not have the resources necessary to buffer periods of increased fetal demands. As such, females who diet and exercise excessively—move ‘too much’ and eat ‘too little’—compromise the development of their fetuses by limiting the availability of calories [6,51]. Therefore, IUGR and SGA are not limited to females in communities with impoverished food environments but may also occur in industrialized nations where maternal concerns about post-natal appearance are prevalent (e.g., weigh gain and striae gravidarum, ‘stretch marks’ [52]).

In sum, inadequate maternal caloric consumption relative to expenditure leads to an impoverished fetal ‘ecosystem’ with fewer cells and reduced metabolic capacity and control. In extreme contexts, negative energy balance in the fetus can result in spontaneous abortion. In less severe cases, IUGR leads to SGA neonates who exhibit lifelong reductions in PA, glucose tolerance, and an increased the risk of T2DM and CVD in adulthood [1,6,10,13].

#### 3.2.2. Atypical IUGR: ‘Big’ Does Not Always Beget ‘Big’

As detailed in Section 4, maternal obesity (excessive fat-cell mass) and inadequate PA play major roles in the ‘over-nourished’ intrauterine environment that leads to ‘large for gestational age’ (LGA) neonates and inherited (pediatric/childhood) obesity [6]. However, it is not uncommon for mothers with obese phenotypes to produce SGA neonates—‘big’ does not always beget ‘big’. As detailed below, this apparent paradox is explained by the interaction of pre-existing *pathologic* IR and hyperplastic obesity.

#### 3.2.3. Maternal Fat-Cells Outcompete Fetal-Cells

Individuals with severe (hyperplastic) obesity—a body mass index (BMI) greater than 40—have an excessive number of fat-cells. As such, the larger population of fat-cells will habitually outcompete all other cells—including those in the fetal ecosystem [1,6]. Normally, this is not a problem for fetal development because the excess adipogenic partitioning and storage reduce the ‘effective caloric intake’ of meals and increase consumption to prevent negative energy balance.

However, if a severely obese female is also sedentary, she may be exhibiting hyperinsulinemia due to the *pathologic* IR induced via physical inactivity. Therefore, the *physiologic* IR of pregnancy may not increase caloric consumption enough to overcome the competition from a large population of maternal fat-cells acting in concert with hyperinsulinemia. In other words, a large population of maternal fat-cells can ‘starve’ fetal-cells of calories and nutrients. This engenders IUGR along a continuum from fetal death to developmentally compromised neonates—as evidenced by the increased prevalence of spontaneous abortion and congenital (structural) abnormalities [53,54,55,56,57,58,59,60].

Importantly, this shows that GWG does not accurately reflect the number of calories captured by the fetus. In other words, in the context of maternal hyperplastic obesity and pre-existing *pathologic* IR, GWG is not a valid indicator of fetal development [61,62]. Recent research supports this hypothesis: *“[i]**n women with pre-pregnancy obesity, GWG seemed to be negatively related to birth weight*…[and]…*“[c]ompared with GWG, pre-pregnancy BMI has greater influence on the birth weight and maternal and infant complications, and may even compromise the benefits of weight control during pregnancy.”* [62].

In sum, maternal hyperplastic obesity in concert with inactivity-induced *pathologic* IR may ‘starve’ a fetus of the calories and nutrients it needs for development.

#### 3.2.4. Sitting-Induced IUGR

In animal research, IUGR can be experimentally produced via ligation of the uteroplacental vessels. This research shows that limiting the supply of oxygen and nutrients to a developing fetus leads to developmental abnormalities such as reduced neurologic, musculoskeletal, and metabolic capacity (e.g., glucose storage) [63].

In humans, maternal obesity in concert with extended periods of sedentary behavior may also lead to IUGR via compression of the uteroplacental vessels. To be precise, higher body mass and sitting increase intra-thoracic, intra-abdominal, and intra-pelvic pressures [64]—and increased pressure correlates with comorbidities [65]. Thus, when the intra-abdominal and intra-pelvic spaces are crowded by excessive fat mass, the act of sitting compresses the uterus, placenta, and supporting vasculature—potentially leading to placental and fetal ischemia. This risk may be similar to that exhibited by pregnant women with heart disease [66].

Although transient sedentary behavior is not problematic, long periods of uninterrupted sitting potentially convey significant risks [24,67,68,69,70]. Importantly, the average pregnant woman spends more than 50% of waking hours sitting—and over 15% of pregnant women spend >5 h/day using sedentary screen-based media in addition to occupation-based sedentary behaviors [71,72]. This volume and frequency of sedentary behaviors represents a potential risk for fetal ischemia in women with obese phenotypes.

## 4. Inherited Obesity

In the previous section, we showed how inadequate caloric consumption relative to PA levels led to IUGR and SGA neonates. In this section, we explain how inadequate PA leads to an over-nourished intrauterine environment, ‘large for gestational age’ (LGA) neonates, and inherited (childhood/pediatric) obesity. However, to understand our use of the term ‘inherited’, it is necessary to grasp the distinction between genetic and nongenetic inheritance.

### 4.1. Nongenetic versus Genetic Inheritance

We have written extensively on how the conflation of the term ‘inherited’ with ‘genetic’ led to confusion about the role of ‘genes’ in obesity and metabolic diseases. Given that a detailed exposition of this work is beyond the scope of this review, we offer an overview below and direct readers to select publications for greater detail [6,10,11,48,49].

To begin, the functional unit in biology and biological inheritance is the cell—not DNA molecules. Therefore, because each cell’s physiology and spatiotemporal context determines gene expression, it is essential in analyses of phenotypic development to distinguish between cellular (nongenetic) inheritance and the two types of genetic inheritance (nuclear and mitochondrial) [6,73,74,75,76]. This distinction is most readily apparent when examining the prenatal development of identical and fraternal twins.

The fundamental difference between monozygotic (identical) and dizygotic (fraternal) twins is inherent in the terminology—identical twins develop from a single cell (a fertilized egg) with a single placenta (usually), whereas fraternal twins develop from two different cells (two fertilized eggs) with two different placentas (always). As such, fraternal twins differ in both cellular and genetic inheritance, and their in utero environments, whereas identical twins do not. Thus, the greater phenotypic disparity of fraternal twins is due to the sum of differences engendered by the gene expression induced by different cells developing within different environments in concert with inter-twin differences in genomes [11].

Therefore, given the variation in the developmental competence (oocyte ‘quality’; e.g., mitochondrial content) of every female’s population of eggs [77,78,79,80,81] and the irreversible impact of the intrauterine environment on developmental trajectories [6,48,49,50], the functional distinctions between cellular, genetic, and environmental (*in utero*) inheritance cannot be ignored.

Consequently, those who infer genetic causality from twin and adoption studies are mistaken because offspring inherit a great deal more than genes from their mothers—they inherit the entire cellular ‘machinery’ in which their genes are expressed. Therefore, the genome is merely an *“organ”* [82] or ‘tool’ of the cell [6,48,49], and a ‘great deal of biology’—both established and undiscovered—links an individual’s nuclear and mitochondrial genotypes, the cellular expression of those genotypes, and phenotypic development.

This suggests that there are no ‘genes for’ quantitative (non-discrete) phenotypes, and that IUGR, stunting, SGA and LGA neonates, and inherited obesity *are caused by* adverse intrauterine environments—independent of genotype [6,10,13,47,48,49,50]. In other words, starve any pregnant mammal and she will abort or bear stunted offspring—independent of her genome.

### 4.2. The Etiology of Inherited (Pediatric/Childhood) Obesity

Inherited obesity is caused by the failure of maternal skeletal muscle-cells to compete with the developing fetal ecosystem and prevent excessive intrauterine positive energy balance [1,6,13]. Specifically, because mammalian pregnancy evolved in prehistoric environments requiring voluminous amounts of bodily movement, appropriate fetal development requires a minimum level of PA to balance the number of calories consumed by the mother and those captured by the maternal and fetal ecosystems.

As detailed earlier and elsewhere (see [1,2,6,13]), PA is essential in maintaining insulin sensitivity and metabolic capacity and control (e.g., storage of blood sugar). Therefore, insufficient PA engenders *pathologic* IR that—like *physiologic* IR—induces hyperinsulinemia, hyperphagia (overconsumption), and positive energy balance. Thus, in the context of pre-existing *pathologic* IR, the *physiologic* IR of pregnancy can exponentially increase maternal caloric consumption while decreasing the competitive advantages of maternal skeletal muscle-cells.

The extreme hyperphagia (overconsumption), in concert with a lack of competition from maternal skeletal muscle-cells, allows the developing fetal ecosystem to capture an excess number of calories. As depicted in Figure 3, this ‘over-nutrition’ stimulates growth in all fetal cells, but given the variability in plasticity, excess caloric capture causes disproportionate increments in pancreatic beta- and fat-cell number (hyperplasia) in concert with dysfunctional skeletal muscle-cell development (more structural and less contractile elements) [6,13].

These pathologic ‘*maternal-effects*’ (non-genetic mechanisms of inheritance [6,7,8,9]) are irreversible and produce offspring who ‘*eat more, store more, and move less*’, independent of genotype. Specifically, the excessive hyperplasia of fetal pancreatic beta-cells induces life-long hyperinsulinemia that, in concert with the larger population of fat-cells, causes excessive adipogenic partitioning—in adipose tissue and ectopically—with subsequent hyperphagia and gains in body and fat mass [1,6,10,13].

Moreover, the dysfunctional development of fetal skeletal muscle-cells causes irreversible reductions in the contractile strength of muscle tissue with concomitant reductions in strength-to-weight ratio. This leads to reductions in PA (weaker individuals move less than stronger individuals) that further reduce the offspring’s metabolic capacity and control.

In sum, these pathologic ‘*maternal-effects*’ engender inherited obesity with life-long decrements in PA and concomitant increments in adipogenic partitioning and metabolic diseases (e.g., T2DM and CVD) [1,6,10,13].

### 4.3. ‘Grand-Maternal Effects’ and Trends in Obesity and T2DM

Our work shows that beginning in the 1950s, there was a substantial decline in the PA of young women and mothers [71,72,83]; see also [84]. Because adequate PA is essential for maintaining the metabolic control necessary for appropriate fetal development, the decades-long decline—in concert with ‘*grand-maternal effects*’—produced multiple generations of females with compromised metabolic phenotypes.

To be precise, ‘*grand-maternal effects*’ occur when female offspring recapitulate the pregnancies of their mothers—phenotype begets phenotype [6,48,49]. Accordingly, these effects impact at least three generations—the grandmother, her children, and her female children’s children. For example, female offspring born to obese, sedentary mothers will ‘*move less, store more, and eat more*’ compared to those born to metabolically healthy mothers. As a result, these females will bear offspring with progressively deteriorating metabolic capacity and control.

Consequently, ‘*grand-maternal effects*’ offer a consilient and comprehensive explanation for the exponential increase in the prevalence of childhood obesity, severe obesity, and adolescent T2DM over the past 50 years—and suggests that future increments are probable. Furthermore, our work suggests that these non-genetic mechanisms of inheritance also explain the increased prevalence of obesity, *pathologic* IR, and T2DM in nonhuman mammals inclusive of dogs, cats, horses, and lab animals [1].

## 5. Summary and Conclusions

During pregnancy, a mother’s cells compete with those of her fetus. To ensure that the developing fetal ecosystem captures the calories it needs for appropriate growth and development, pregnancy induces *physiologic* IR that decreases the competitive advantages of maternal skeletal muscle-cells and increases maternal caloric consumption. This leads to the positive energy balance required for fetal development.

Importantly, because mammalian pregnancy evolved in environments that necessitated voluminous amounts of PA, appropriate fetal development requires a minimum level of PA to balance the number of calories captured by the maternal and fetal ecosystems. Although the developing fetal ecosystem can withstand mild perturbations, excessive maternal PA relative to caloric consumption or inadequate maternal PA with concomitant *pathologic* IR negatively—and irreversibly—alter offspring metabolic phenotypes and health trajectories.

Specifically, excessive PA relative to caloric consumption induces a negative intrauterine energy balance that engenders an impoverished fetal ecosystem (SGA neonates) with fewer skeletal muscle-cells and concomitant reduced metabolic capacity and control. This IUGR leads to an increased risk of acquired (adult-onset) obesity, T2DM, and CVD as posited by the ‘*Barker Hypothesis*’ [43,44,45,46] and the more comprehensive ‘*Maternal Resources Hypothesis*’ [6,10,47,48,49,50].

Conversely, inadequate maternal PA induces *pathologic* IR that—in concert with the *physiologic* IR of pregnancy—exponentially increases the number of calories captured by the fetal ecosystem. This ‘over-nourishment’ causes LGA neonates with disproportionate growth in fetal pancreatic beta- and fat-cells in concert with dysfunctional skeletal muscle-cell development. The larger population of fat-cells acting in concert with chronic hyperinsulinemia exponentially increases the number of calories captured, stored, and retained in fat-cells after each meal. This extreme adipogenic partitioning reduces the ‘effective caloric intake’ of meals, thereby causing overconsumption and gains in body and fat mass. Consequently, the offspring of physically inactive, metabolically compromised mothers will ‘*move less, store more, and eat more*’ throughout their lives.

Importantly, female offspring born to obese, sedentary mothers will recapitulate the pregnancy of their mothers, leading to progressive decrements in metabolic capacity and control across generations. As such, the inheritance of metabolic phenotypes via inactivity-induced ‘*maternal-*’ and ‘*grand-maternal effects*’ offers a consilient and comprehensive explanation for much of the increase in obesity and T2DM over the past 50 years in human and non-human mammals [1,2,6,10,13,50].

## Figures and Tables

**Figure 1 metabolites-13-00545-f001:**
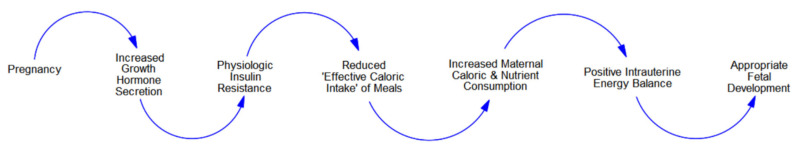
Normal pregnancy.

**Figure 2 metabolites-13-00545-f002:**
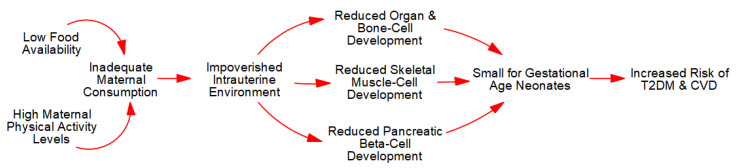
Intrauterine growth restriction.

**Figure 3 metabolites-13-00545-f003:**
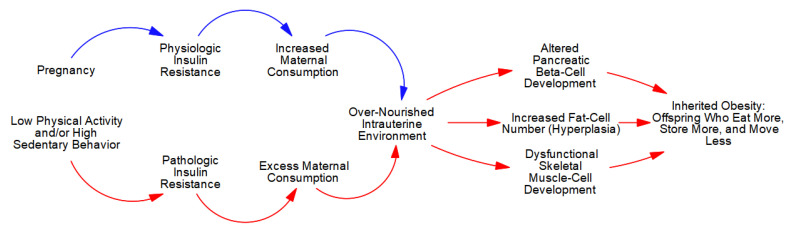
Inherited obesity.

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
