# Peer review of "Metabolic Inheritance and the Competition for Calories between Mother and Fetus"

_metabolites, 2023, doi:10.3390/metabo13040545_

Round 1

Reviewer 1 Report

My overall opinion on the assessed work entitled: “Metabolic Inheritance and the Competition for Calories between Mother and Fetus” is positive.

I consider the manuscript of paper submitted for evaluation in Journal “Metabolites” to be a potentially valuable manuscript, properly planned as well as worthy of accepting and publishing only in case of implementation of the following corrections.

Remarks:

The Authors of the manuscript text, omitted the justification for the scientific task of a specified in title i.e. "Metabolic Inheritance and the Competition for Calories Between Mother and Fetus".

It is also difficult to find a clearly defined aim of the submitted manuscript.

The Introduction is laconic and too general.

The Authors do not specify how the "Metabolic Inheritance and The Competition for Calories Between Mother and Fetus" affects activity of the key the secretory organ of the human body – adipose tissue cells?

The Authors did not discuss how "Metabolic Inheritance and The Competition for Calories Between Mother and Fetus" affects the activity of insulin receptors.

The Authors did not discuss how "Metabolic Inheritance and The Competition for Calories Between Mother and Fetus" affects the activity and expression of glucose transporters.

Relevant fragments of the text are devoid of literature (citations) references, i.e. lines 101-126.; lines 136-142.; lines 154-175; lines 177-187.; lines 194-200.; lines 211-223.; lines 315-323;

The fonts in figures and in the text are different.

The year of publishing of selected literature items (no. 10., no. 17., no. 19., no. 34., no. 45., no. 71.,) was not thickened.

Author Response

Dear Editors and Reviewers,

We appreciate the detailed review and revised our manuscript accordingly. Our responses (below) are denoted by a red font, and revisions in the manuscript are denoted via track-changes.

It is important to note that — as per our invitation — the submitted manuscript is not presenting a novel hypothesis but is an overview of our previously published — and highly cited — work. Our scholarship in this domain includes dozens of papers and book chapters with thousands of references for scientific support. In addition, our theories were the basis for hundreds of popular media articles in prestigious magazines and newspapers including The New York Times, as well as articles written by our lead author in the ‘New Scientist’, ‘Real Clear Science’, ‘The Conversation’ and ‘The Washington Post’.

Therefore, given the scope of our prior work and theories, it would be impossible — in a single paper — to provide details of the myriad mechanisms subsumed by our ideas. As such, in both the original and revised manuscript, we included the phrase “As detailed elsewhere…” to direct readers seeking greater scientific support to the relevant references.

Reviewer #1

My overall opinion on the assessed work entitled: “Metabolic Inheritance and the Competition for Calories between Mother and Fetus” is positive.

I consider the manuscript of paper submitted for evaluation in Journal “Metabolites” to be a potentially valuable manuscript, properly planned as well as worthy of accepting and publishing only in case of implementation of the following corrections.

Remarks:

The Authors of the manuscript text, omitted the justification for the scientific task of a specified in title i.e. "Metabolic Inheritance and the Competition for Calories Between Mother and Fetus".

Our response: As stated in our Abstract (lines 21-22 of the revised manuscript), the justification for our paper was offering “a comprehensive and consilient explanation for much of the increase in obesity and T2DM over the past 50 years in human and non-human mammals.”

Similarly, as per our Introduction (lines 38-43 of the revised manuscript), we presented “a comprehensive framework for understanding the inheritance of metabolic phenotypes and subsequent development of obesity, type-2 diabetes mellitus (T2DM), and cardiovascular diseases (CVD) in human and nonhuman mammals.”

Furthermore, we stated in our cover letter that our paper provides a concise yet comprehensive overview of the inheritance of metabolic phenotypes.” 

We look forward to the reviewer’s suggestions on how we can express these ideas more clearly.

It is also difficult to find a clearly defined aim of the submitted manuscript.

Our response: As we wrote on lines 37-43 of the revised manuscript, this paper “offers a comprehensive framework for understanding the inheritance of metabolic phenotypes and subsequent development of obesity, type-2 diabetes mellitus (T2DM), and cardiovascular diseases (CVD) in human and nonhuman mammals….[and explains]… how the ‘competition for calories’ between mother and fetus impacts inheritance across species...”. 

We look forward to the reviewer’s suggestions on how we can express these ideas more clearly.

The Introduction is laconic and too general.

Our response: Our theories are broad and consilient. Therefore, it is necessary to present the requisite background information prior to the details of the specific relations and mechanisms. In other words, a brief, general introduction will avoid confusing the reader. 

The Authors do not specify how the "Metabolic Inheritance and The Competition for Calories Between Mother and Fetus" affects activity of the key the secretory organ of the human body – adipose tissue cells?

Our response: Adipose tissue consists of many cell types such as mesenchymal stem-cells, pre-adipocytes, and white, brown, and beige adipocytes. Nevertheless, a discussion of each of these cell-types is well-beyond the scope of our discussions of fat-cell (adipocyte) metabolism and its impact on the inheritance of metabolic phenotypes.

Nevertheless, given that some readers may not be aware that the term ‘fat-cell’ is synonymous with the term ‘adipocyte’, we revised our text to make this explicit (please see line 109-110 of the revised manuscript).

Importantly, we used the term ‘fat-cell’ ~40 times throughout the original manuscript and our subheading “Fat-Cells” (lines 173 to 196 of the revised manuscript) describes the reciprocal relation between fat-cell number, the ‘effective caloric intake’ of meals, and gains in body and fat mass. In lines 195-196 of the revised manuscript we wrote that “maternal insulin sensitivity, fat mass, and fetal fat-cell hyperplasia play decisive roles in the inheritance of metabolic phenotypes.” And in lines 407-417 of the revised manuscript we specified how ‘maternal-effects’ engender fat-cell hyperplasia and the adipogenic partitioning of calories.   

The Authors did not discuss how "Metabolic Inheritance and The Competition for Calories Between Mother and Fetus" affects the activity of insulin receptors.

Our response: We made the assumption that our readers understand that insulin receptors play a key role in insulin sensitivity, insulin resistance (IR), and subsequent glucose metabolism. However, the level of detail necessary to capture the activity of insulin receptors per se is well-beyond the scope of the present manuscript — especially since we are presenting an overview of our previously published work that provides that level of detail (please see references 1, 2, 6, 10, 13, 19, 24 in the revised manuscript). These works offer meticulous detail on how the competition for calories affects, and is in turn affected by, contraction-mediated (non-insulin) and insulin-mediated mechanisms and their concomitant effects on insulin receptors and glucose metabolism.

The Authors did not discuss how "Metabolic Inheritance and The Competition for Calories Between Mother and Fetus" affects the activity and expression of glucose transporters.

Our response: As per our previous response, insulin sensitivity and IR are predicated on the responsiveness of glucose transporters to contraction-mediated (non-insulin) and insulin-mediated mechanisms. We cited our previously published work that provides meticulous detail on these mechanisms. Please see references 1, 2, 6, 10, 13, 19, 24 in the revised manuscript. 

However, this level of detail is well-beyond the scope of the present paper — especially since it is an overview of our previously published work.

Relevant fragments of the text are devoid of literature (citations) references, i.e. lines 101-126.; lines 136-142.; lines 154-175; lines 177-187.; lines 194-200.; lines 211-223.; lines 315-323;

Our response: We appreciate this critique and revised our manuscript by adding the requisite citations.

The fonts in figures and in the text are different.

Our response: Yes, it is common to use different fonts to draw attention to figures. We used this technique in hundreds of our publications.

The year of publishing of selected literature items (no. 10., no. 17., no. 19., no. 34., no. 45., no. 71.,) was not thickened.

We thank the reviewer for the careful reading and revised our paper accordingly.

Reviewer 2 Report

The manuscript “Metabolic Inheritance and Competition for Calories between Mother and Fetus” suggests potential problems of metabolic inheritance between the mother and fetus. The authors approached and concluded these problems from a different point of view using the word “competition”. It is an interesting interpretation, but it feels more like an essay. It is because the contents are lacking interpretation based on scientific facts.

Some points have to be corrected.

Major points

1. In line 68, It is hard to understand “competition for calories”. Is “competition for calories” necessary to live?

2. In line 93, I do not agree with the author’s interpretation. I think that cells also change depending on the individuality, behavior, and intake balance of humans and animals.

3. In line 148, the authors explain adipogenic partitioning about fat cells. Is this partitioning not seen in other cells?

4. In lines 167-173, the contents are lacking scientific facts. Please add related references.

5. In lines 342-343, I understand somewhat. What are the criteria for non-obesity? The criteria for the success of maternal skeletal muscle cells is ambiguous.  

Minor points

1. In line 18, Please check the word size of the “leading”.

2. In lines 291-292, Please check the sentence.

3. In line 298, Please add “,” after “section”.

4. In line 310, please amend “spaciotemporal” to “spatiotemporal”

Author Response

Dear Editors and Reviewers,

We appreciate the detailed review and revised our manuscript accordingly. Our responses (below) are denoted by a red font, and revisions in the manuscript are denoted via track-changes.

It is important to note that — as per our invitation — the submitted manuscript is not presenting a novel hypothesis but is an overview of our previously published — and highly cited — work. Our scholarship in this domain includes dozens of papers and book chapters with thousands of references for scientific support. In addition, our theories were the basis for hundreds of popular media articles in prestigious magazines and newspapers including The New York Times, as well as articles written by our lead author in the ‘New Scientist’, ‘Real Clear Science’, ‘The Conversation’ and ‘The Washington Post’.

Therefore, given the scope of our prior work and theories, it would be impossible — in a single paper — to provide details of the myriad mechanisms subsumed by our ideas. As such, in both the original and revised manuscript, we included the phrase “As detailed elsewhere…” to direct readers seeking greater scientific support to the relevant references.

Reviewer #2:

The manuscript “Metabolic Inheritance and Competition for Calories between Mother and Fetus” suggests potential problems of metabolic inheritance between the mother and fetus. The authors approached and concluded these problems from a different point of view using the word “competition”. It is an interesting interpretation, but it feels more like an essay. It is because the contents are lacking interpretation based on scientific facts.

Some points have to be corrected.

Major points

  1. In line 68, It is hard to understand “competition for calories”. Is “competition for calories” necessary to live?

Our response: We are open to suggestions on how to improve the clarity of our statements. We cited Darwin’s ‘Origin of the Species’ because ‘evolution by natural selection’ is predicated on the competition for resources and, as we wrote on lines 61-62 of the revised manuscript, ‘calories’ are a finite resource.

Therefore, to understand our use of the term “competition for calories” , it is necessary to grasp how it was presented in the previous sentences. For example, in lines 34-36 we wrote, “mammalian bodies can be viewed as ‘ecosystems’ comprised of trillions of semi-autonomous cells competing for the nutrient-energy resources necessary for survival (‘calories’; glucose, fatty- and amino-acid molecules).”  

In lines 55-57, we wrote “because all cells require nutrient-energy, the fundamental ‘problem’ in evolution was how a cell could acquire the ‘calories’ it needed for survival and reproduction.”

And in lines 63-64 of the revised manuscript, we wrote “differences in cell phenotypes (physiology and behavior) led to differences in caloric acquisition with concomitant differential survival and reproduction.”

These sentences explicitly state that ‘calories’ are finite resource that is necessary for survival, and that the ‘competition for calories’ led to differential survival. 

Importantly, we made the assumption that our readers would understand that ‘to survive’ is synonymous with ‘to live’. We cannot replace “survival and reproduction” with “to live and reproduce”, because that phrasing is not precise nor acceptable in evolutionary biology.

Nevertheless, we are open to other suggestions on how to improve the clarity of our statements.

  1. In line 93, I do not agree with the author’s interpretation. I think that cells also change depending on the individuality, behavior, and intake balance of humans and animals.

Our response: We appreciate the reviewer’s lack of agreement but respectfully note that the mechanisms we present are predicated on how “individuality” and “behavior” determine the balance between caloric intake and expenditure. In other words, our interpretation anticipates and subsumes the reviewer’s perspective. 

Therefore, the reviewer’s disagreement may be due to a lack of familiarity with the details of our prior work as cited throughout the manuscript. For example, please see references #1 & 2 in the revised manuscript (below):

 Archer et al. Cell-Specific "Competition for Calories" Drives Asymmetric Nutrient-Energy Partitioning, Obesity, and Metabolic Diseases in Human and Non-human Animals. Front Physiol 2018;9(1053):1053. doi: 10.3389/fphys.2018.01053 [published Online First: 2018/08/28] 

And

Archer & Hill. Body and fat mass are not regulated, controlled, or defended: An introduction to the ‘invisible Hand’ and ‘competition’ models of metabolism. Progress in Cardiovascular Diseases 2022 doi: https://doi.org/10.1016/j.pcad.2022.10.003

These cited works explain — in meticulous detail — how individuality and behavior determine the balance between caloric intake and expenditure.

  1. In line 148, the authors explain adipogenic partitioning about fat cells. Is this partitioning not seen in other cells?

Our response: The partitioning of ‘calories’ (nutrient-energy partitioning) occurs as a result of the competition between all cell types — which in turn is based on cell-type-specific competitive behaviors (please see lines 86-87 and 127-130 of the revised manuscript). So yes, partitioning occurs to all cells but “adipogenic partitioning” (by definition) is specific to fat-cells (adipocytes).

To be precise, we explain the partitioning of calories to skeletal muscle-cells in lines 133-170 of the revised manuscript. However, we failed to make this clear by using terms related to partitioning such as “acquisition”, “capture”, and “exploit”.

Accordingly, we revised our manuscript to correct the ambiguity. Please see lines 150-151 of the revised manuscript. 

  1. In lines 167-173, the contents are lacking scientific facts. Please add related references.

Our response: We thank the reviewer for the close reading and revised accordingly with several new references (please see lines 183, 186, 189, and 194 of the revised manuscript).  

  1. In lines 342-343, I understand somewhat. What are the criteria for non-obesity? The criteria for the success of maternal skeletal muscle cells is ambiguous.

Our response: We edited lines 393-394 in the revised manuscript to clarify the competition between maternal skeletal muscle-cells and the fetal ecosystem: “Inherited obesity is caused by the failure of maternal skeletal muscle-cells to compete with the developing fetal ecosystem and prevent excessive intrauterine positive energy balance.”

Additionally, with respect to ‘normal’ (non-obese pregnancies), we revised lines 396-397, “appropriate fetal development requires a minimum level of PA to balance the number of calories consumed by the mother and those captured by the maternal and fetal ecosystems.”

In other words, all pregnancies require a minimum level of maternal PA to ensure adequate — but not excessive — maternal consumption and sufficient competition between maternal skeletal muscle-cells and the fetal ecosystem.

Minor points

  1. In line 18, Please check the word size of the “leading”.

We appreciate the close reading and corrected the error.

  1. In lines 291-292, Please check the sentence.

We appreciate the close reading and revised the sentence (line 339 of the revised manuscript): “Although transient sedentary behavior is not problematic, long periods of uninterrupted sitting potentially convey significant risks.”

  1. In line 298, Please add “,” after “section”.

We appreciate the close reading and added the requisite comma.

  1. In line 310, please amend “spaciotemporal” to “spatiotemporal”

We appreciate the close reading. Although “spaciotemporal” is synonymous with “spatiotemporal”, we revised the manuscript to the latter.

Round 2

Reviewer 1 Report

The Authors of the publication entitled "Metabolic Inheritance and the Competition for Calories between Mother and Fetus" modified the manuscript - at the same time referring to the comments of the reviewers. 

The article in its current form is suitable for publication without corrections.

Reviewer 2 Report

I think that the revised manuscript has been fundamentally improved and that it includes the contents requested by the referees and editorial team.